# The Inter/Transdisciplinary Framework for Urban Governance Intervention in the Egyptian Informal Settlements

Eman El Nachar [1,2] and Doaa Abouelmagd [2,*]

1    Postgraduate Faculty of Interdisciplinary Studies and Research, Helwan University, Ain Helwan, Cairo 11795, Egypt
2    Architecture Department, Faculty of Fine Arts, Helwan University, Zamalek, Cairo 11561, Egypt
*    Correspondence: dabouelmagd@f-arts.helwan.edu.eg

**Abstract:** There is a need to understand the complex nature of informal settlements to achieve Sustainable Development Goal 11: "*Making cities and human settlements inclusive, safe, resilient, and sustainable*". Thus, addressing the urban governance of informal settlements requires an inter/trans-disciplinary scope to reach a cross-cutting agenda that combines social/behavioral, economics, and public health with the built environment disciplines. Respectively, this paper aims to establish an integrative framework based upon a blended inter/trans-disciplinary approach of urban governance for informal settlements in Egypt. The study adopts a theoretical analytic methodology to achieve its aim by reviewing the literature on informal settlement policies. It argues that the inter/trans-disciplinarity approach contributes to integrative urban governance agendas that enhance the quality of life of informal settlements. While exploring three bottom-up perspectives of understanding the formation of informal urban settings—socioeconomic, morphological, and sociophysical—an integrative model is developed to allow a contextual perspective for examining informal domains. The model is articulated to guide the purpose of the multidimensional analysis methods for investigating informal contexts. An integrative agenda with six analysis tasks, each involving interdisciplinary group of academics, experts, stakeholders, and authorized representatives, is outlined by the method originated in this paper. Ultimately, and concerning aspects of sustainable urban cities, the paper introduces an integrative agenda that enables overcoming gaps in current upgrading practices when examining the informal settlements of the Egyptian contexts.

**Keywords:** urban governance; interdisciplinary/transdisciplinary studies; informal settlements; Egypt





## 1. Introduction

In 2018, more than four billion inhabitants lived in urban areas, while 828 million were living in slums; in 2050, those numbers are expected to have increased by 155%. This rapid urbanization is connected to migration and urban expansion in megacities. The developing world is expected to house nine of each ten megacities in the future. Slums and informal settlements become a significant part of the cities where the low and middle classes live and work, and they present the majority of the workforce that generates 80% of the world's GDP [1]. This means cities in developed and developing countries face economic, environmental, social, and political challenges caused by the uncontrollable rapid expansion of urban areas at both spatial and demographic levels. Despite all the housing and urban development policies that have emerged since WWII, failures have been witnessed together with social, economic, and other 'wide gaps' among different interests involved in complex problems generated by the unplanned urban fabric. Although most efforts were directed at achieving economic development, low-income groups remain on the margin while mostly neglected in formal spatial planning and developing relevant policies. In essence, they are increasingly pushed into informality, especially regarding housing demands and public infrastructure and services needed for survival [2,3].

In accordance, cities have become the focus of development, and slum dwellers have become the focus of safe and resilient sustainable development [1,4]. In 2015, the release of the United Nations SDGs, followed by the Habitat III conference in 2016, provided a shift toward the sustainable integrated urban approach. It includes a sustainable approach to developing and empowering slum dwellers, integrating equity and social justice into urban development.

Egypt's 2030 strategic plan for sustainable urban development emanating from SDGs has raised serious agendas to recover the quality of life in the informal contexts all over Egypt through economic, social, and environmental dimensions. Moreover, urban development through housing projects and in situ development has become a pillar of the state's development agenda to improve the quality of Egyptian lives [5]. The term informal settlement is used by the Egyptian state when dealing with slums that are categorized based on existing physical and legal conditions of the buildings to unplanned and unsafe areas [6]. Egypt's SDG indicator's performance shows Egypt faces significant challenges to achieving the goals of sustainable cities and community livability. Although there has been a decrease in the percentage of inhabitants living in informal settlements, challenges remain [7]. In response, this paper questions how urban governance agendas could allocate complexity and multi-facets of informal contexts to improve residents' quality of life. The fundamental premise of the paper is to consider informal practices of public intervention in unplanned contexts as sources of networked forms of power that work beyond the formal government and many other actors involved, including authoritarians and experts of social, economic, urban, and health domains. This paper claims that an innovative interdisciplinary/ transdisciplinary vision is the key approach to effective urban governance plans for unplanned settlements in Egypt. The complexity and dynamic nature of the informal settlements cannot be examined from a single discipline; it requires the interrelation between diverse disciplines to understand this challenging problem with long-term impact. It also involves the local community and all related stakeholders, including the private sector and NGOs.

Few pieces of literature have called for a holistic approach to dealing with informal settlements that go beyond one single approach and combine the socioeconomic, physical, morphological, and environmental constructs [8–10]. As a continuity of the holistic vision to understand and examine the slums with an interdisciplinary approach, this paper presents a novel operational and procedural inter/transdisciplinary framework in Egypt.

The research argument underlines how engaging of socio/spatial discourse is essential to efficient urban sustainable strategies for the informal contexts. Accordingly, examining informal areas through interdisciplinary/transdisciplinary approaches provides an integrative knowledge base for urban governance agendas that improve qualities of life in informal urban settlements.

Based upon a contextual perspective, the paper aims to introduce a comprehensive conceptual agenda for examining informal settlements in Egypt. The agenda is intended to develop efficient urban governance strategies that reduce current gaps and conflicts among different perspectives and resources when dealing with informal urban communities. It invites different domains of expertise and stakeholders to practice inter/transdisciplinarity by introducing an integrative framework for examining informal communities.

Merging sociocultural dimensions, the study adopts a theoretical analytic methodology to achieve its aim. Reviewing the literature on informal settlement policies and examinations provided insights into several shortcomings.

The theoretical analyses also justify why the inter/transdisciplinary approach contributes to integrative urban governance agendas that enhance the quality of life of informal settlements; at the same time, exploring three bottom-up perspectives of understanding the formation of informal urban settings: socioeconomic, morphological, and sociophysical. A three-layer analytic framework inspired by Salama's work on the Lefebvrian triadic conception and the production of space is settled to understand the formation process of informal urban settlements [11]. An integrative agenda for examining informal settlements

is outlined in six analysis tasks, each performed by interdisciplinary groups of academics, experts, stakeholders, and authorized representatives.

Examination undertakings include political economy analysis, socioeconomic network analysis, and socio/spatial analysis. While tackling issues such as residents' satisfaction, identity, attachment, attitudes, and preferences towards the built context and the potential improvements, we provide a comprehensive knowledge base that could lead to interventionist urban governance strategies for the unplanned areas in Egypt.

## 2. Literature Review and Theoretical Background

The literature review conducted in this paper is intended to explore three key issues that could establish the theoretical foundation to achieve the research aim.

- Firstly, while identifying common qualitative and qualitative aspects of the Egyptian informal contexts, current shortcomings and gaps in practices of policies and approaches to upgrading informal settlements are identified.
- Secondly, driven by the complex problems of the informal contexts, the question of how to develop agendas for integrative urban governance is central. Thus, literature in the field was scanned to ascertain how the interdisciplinary and transdisciplinary research approaches are essential to informal contexts' examinations.
- Thirdly, an overview of the bottom-up approaches to understanding the mechanism of informal settlement formation is explored. Ultimately, an integrative model is articulated to guide multidimensional analysis procedures for investigating informal contexts. Following are expanded discussions of the aforementioned three key issues.

### 2.1. Allocating Informal Settlements in Egypt—Current Gaps

Since the 1960s, several strategies and policies worldwide have tackled the problems of informal urbanization through various approaches that range from the eradication of informal settlements and rehousing the people, most likely in public or social housing, to establishing programs focusing on tenure legalization, infrastructure improvements, and facilitation of credit to encourage self-help housing and socioeconomic development [12–14].

In the 1960s, informal settlements were mostly perceived as obstacles to building modern cities, and the slum clearance agenda was massively adopted; in this phase, the governments applied direct housing provisions. The overall idea was eradicating informal contexts from urban areas with government-led capital-intensive housing facilities. The 1970s–1990s marked the shift towards the support-based approach in Turner's work in Latin America and Habitat I conference. In this phase, several pilot projects took place with community participation; examples can be seen worldwide in site and service, slum upgrading, and other self-help models adapted with the support of the World Bank. Nevertheless, the level of community participation remained small and limited. Afterward, shifts towards more integration into citywide policies and institutional reforms were emphasized to provide infrastructure improvements, social services, and physical restructuring of informal settlements, followed by the legalization of tenure and regularization of property rights [15,16].

In 1987, "the global strategy for shelter for the year 2000" was released and paved the shift toward the enabling approach. This shift was ensured with the Habitat II conference in 1996 and the Millennium Development Goals (MDGs) in 2000. This phase witnessed governments shift from the role of the provider to the enabler and the participation of different actors, including local communities, the private sector, and NGOs. With all these policy shifts, the gap between the housing market and the informal sector kept increasing with its complex problems.

Subsequently, the definition of governance was introduced during the 1990s to designate the authority's administrative capabilities to manage a country's affairs at all levels. It comprises the mechanisms, processes, and institutions through which citizens and groups "articulate their interests, exercise their legal rights, meet their obligations, and mediate their differences" [17]. Different actors' voices should be represented to achieve a successful

governance process, even those of the most vulnerable groups [18]. Accordingly, the United Nations Centre for Human Settlements (UNCHS) proposed a set of indicators to measure the success of governance within the urban context where effective governance would act as a trigger (cause) for better urban performance (effect).

From this perspective, the term urban governance includes networked forms of power that work beyond the formal government and involve many other actors in procedures that determine how urban development evolves to meet the needs of several groups' interests and growth expectations.

However, the available body of knowledge in the field reveals enormous gaps between the logic of governance and survival [19]. Consequently, marginalized groups mobilize to fulfill their missing needs [20]. Urban development and public services, including infrastructure, result from government intervention through formal and informal societal processes [21,22]. Conflict and overlapping of formal and informal governance systems increasingly constitute a critical and legitimate component of city development [23].

On the national level, despite the shifts in housing policies at the international level, the Egyptian state kept playing the role of housing provider for the low-income groups, which created a gap in the housing market between what was provided and the real need of the low-income groups. Most of the projects were provided under the schema of the new cities, but the low-income groups left them in favor of the informal settlements where they could find a livelihood. Between the 1970s and 1990s, 80% of the housing stock in Greater Cairo was built in informal areas [24]. Pilot projects were introduced under the site and services schema showing a failure in applying this policy due to the incomplete process of providing infrastructure and project location. In 2014, due to the informal boom after the 2011 revolution, 60–70% of the urban fabric in Egypt was considered informal; however, severe governmental strategies tackled the challenges of urban informality. By the beginning of the 1990s, a systematic approach was initiated to upgrade informal settlements throughout Egypt. In 1993, the Egyptian state founded the national fund for urban upgrading [24]. A national survey to identify informal settlements was carried out in urban areas [25].

Consequently, a massive informal settlement upgrading program was initiated, and a national plan was developed. The program included two main stages. First stage: 1994–2004 Informal Settlements Development Program (ISDP), mainly aimed at providing infrastructural and essential services for informal settlements and developing deteriorated areas. Second stage: 2004–2008 informal settlement belting program (Tahzeem El-Ashwa'iyyat). In this stage, strategies were focused only on providing infrastructure and improving the physical condition of the deprived areas, while socioeconomic aspects were ignored entirely [25]. At the same time, the concepts of community participation in planning or implementation, the legalization of properties, and the security of tenure were overlooked.

In 2008, the Informal Settlement Development Fund (ISDF) was established (now known as the Urban Development Fund) to coordinate efforts and finance the development of an informal areas program. ISDF has achieved significant change in the ideology of developing informal urbanization by categorizing two types of informal settlements in Egypt, which are the "unsafe areas" and the "unplanned areas", that grew away from legal schemes [6]. This classification includes informal settlements on agricultural land, desert land, historic core, and deteriorated urban pockets [26]. Although informal settlements are different in their typology, population density, crowding rates, and housing conditions, generally, most of them suffer from high population density, high floor area ratio, and lack or absence of economic and social infrastructure services [27].

Typically, Egyptian government strategies towards upgrading the informal settlements have followed two different approaches:

- The preventing approach aims to limit the growth of the current informal settlements and apply various policies, regulations, and tools, including the belting program, to bond the informal areas' borders and prevent building in slums;

- The intervention approach aimed towards improving the current situation of existing informal settlements through authorities such as ISDF, Ministry of Housing, Utilities and Urban Communities (MHUUC), and local non-governmental organizations. Interventions could have many forms, such as resettlement or relocation, improvements in resettlement, rehousing, and upgrades/ rehabilitation [6].

Consequently, several strategies have been followed prioritizing intervention in unsafe areas according to degrees of unsafe conditions to improve shelter complaints. As for unplanned areas, approaches have relied on market-based mechanisms with partnerships of the private sector, residents, and the public sector [6,28].

In 2021, Egypt announced ending the unsafe areas after developing 357 areas in 25 governorates. Consequently, the unplanned classification represents around 60% of the informal settlements asserted to be finished by 2030 in sync with the SDGs and Egypt's 2030 plan [28,29]. ISDF's policy emphasized that whenever applicable, in situ upgrading should be the norm when dealing with informal settlements; 37.5% of the Egyptian cities are classified as unplanned areas, with a total informal area of 152,000 Fadden (Fadden is an area unit equivalent to 1.038 acres) and an estimated population of 22 million. At the end of 2021, 4616 Fadden had been developed, and 6941 were under development [30]. In the metropolitan area of Greater Cairo (Cairo, Giza, and part of the Qalibeya governorate), 9.2 million inhabitants are estimated to live in the unplanned areas, in an area of 22.535 Fadden (see Figure 1 for Greater Cairo unplanned areas, and Figures 2 and 3 for examples of informal settlements on agriculture land [31]).

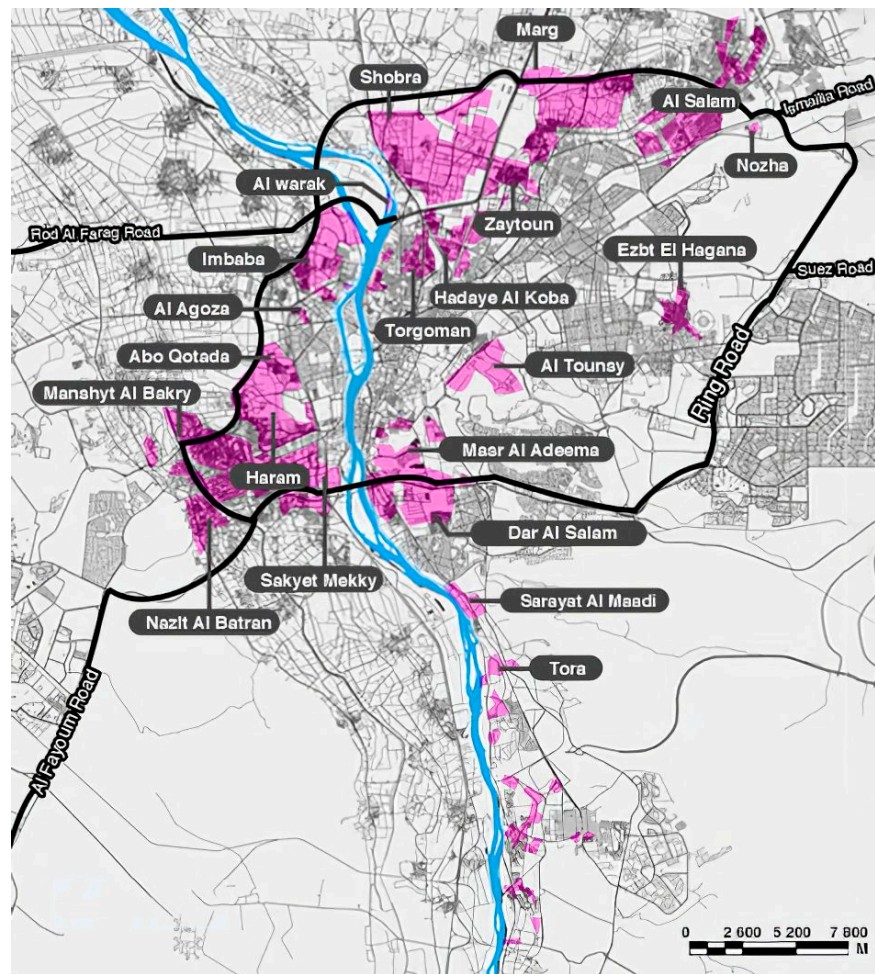

**Figure 1.** Greater Cairo unplanned areas. Map drawn by the authors based on data collected from [31], the base map for Greater Cairo is adapted from dreamstime.com with royalty-free licenses permission.

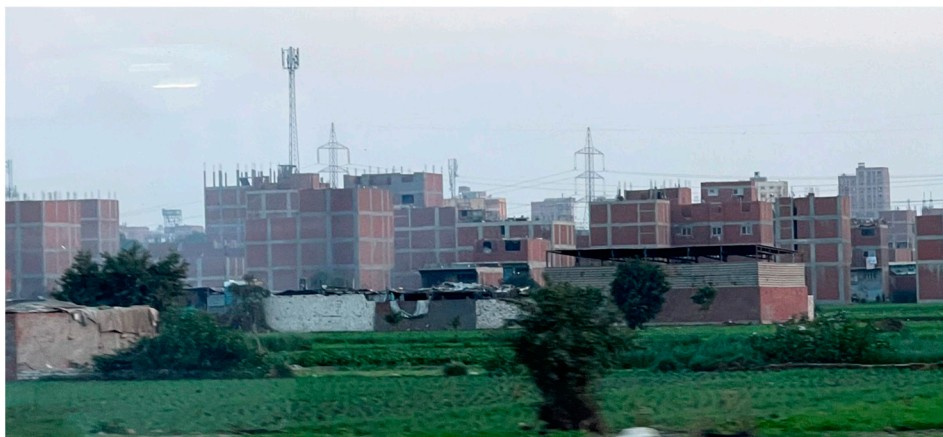

**Figure 2.** Unplanned informal settlement on agricultural lands, Bragil, Greater Cairo periphery. Taken by the authors (2022).

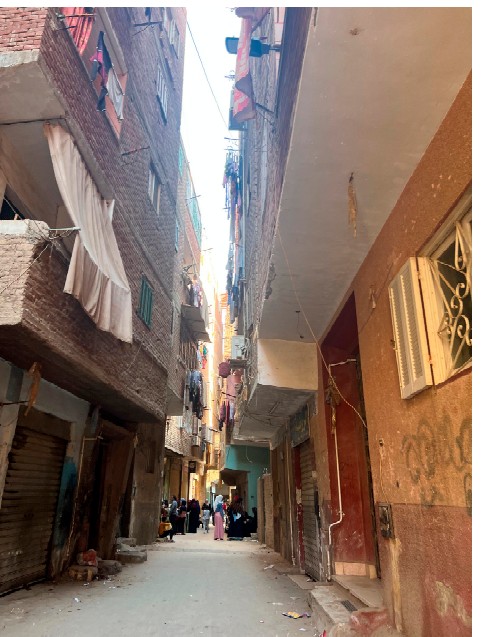 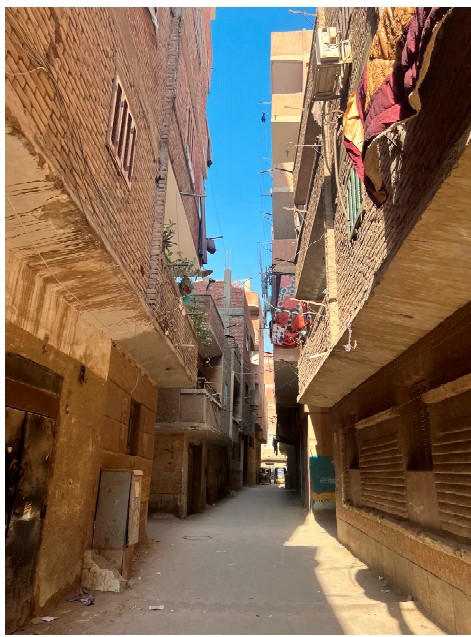

**Figure 3.** The narrow streets with a high density and high floor area ratio of the unplanned informal area, Marg, Greater Cairo. Taken by the authors (2022).

A wide range of literature has discussed approaches, strategies, and experiences to improve life quality in unplanned areas [11,32,33]. Key aspects of managing informal settlements in the Egyptian contexts could be concluded as follows.

Many policies and laws have partially controlled some informal settlements' growth, but they are still growing. At the same time, most intervention projects face many problems and complications in the planning and implementation phases due to conflicting perspectives among stakeholders, whether the government or slum dwellers; the process is perceived as an injustice since it does not serve subjective interests. It also revealed the need for coordination and integration of efforts and objectives, together with the unclear identification of roles for each partner involved in upgrading the areas [6,29].

Understanding the complexity of informal contexts calls for more effective participatory strategies and procedures to ensure better accommodate residents' attitudes, preferences, and style of life in the relocation process and to consider residents' involvement in planning and design phases.

When dealing with the informal contexts, examining the interrelation between qualities of the built environment and sociocultural aspects demands more successful urban

governance agendas. While exploring critical issues of community livability, satisfaction, place identity, and place making are underlined as indicators of quality of life in unplanned contexts [34,35]. In addition, successful experiences revealed that upgrading plans do not work if people are uprooted and lose their source of income and social networks [33,36,37].

From this standpoint, informal urban governance should be perceived as a critical building block in facilitating a bottom-up approach that allows citizens to participate in decision-making processes and improves their urban quality of life [14,38,39].

Concerning urban governance of the unplanned areas in Egypt, the previous review reveals several shortcomings and gaps that could be explained as follows.

- Networking is required for community-building processes to develop relationships and trust among formal and informal governance systems.
- An integrative knowledge base outcome is absent, although informal settlements are studied from multiple dimensions, primarily social, economic, and physical. Improving the sociophysical as well as socioeconomic conditions of the informal contexts calls for developing comprehensive urban governance agendas.
- There needs to be more understanding of informal settlements via multifunctional contexts. Hence, the success of sustainable development agendas at the urban level demands recognizing and supporting this multifunctionality and working on all facets related to improving quality of life, including public health, safety, economic diversity, and community well-being and satisfaction.
- There is a lack of exploration of the deformation stages and processes of the unplanned areas and the actors involved in establishing informal contexts. However, urban governance agendas consider the interrelations of socioeconomic and sociophysical aspects over time.

Based on the above shortcomings, developing and practicing effective urban governance agendas for informal settings to consider sustainable urban development is determined by two main related concerns. First, the multiple facets of sustainable urban development call for interdisciplinary approaches to examine the unplanned areas within their interrelated social, economic, and physical dimensions while generating integrative knowledge base outcomes. Second, understanding the mechanisms of informal contexts' formalization in any urban area is central to developing aspects of effective strategies for urban governance.

### 2.2. Interdisciplinary/Transdisciplinary Approaches for Integrated Urban Governance

A dramatic revolution has been witnessed in understanding policies that address upgrading urban informal settlements in our cities. Moving from government to governance is central to establish more efficent urban politics. New governance forms have gained importance by involving civil society (NGOs, businesses, residents) in making and implementing decisions. Two main themes frame the urban governance agenda for the new millennium. The first recognizes the importance of participation of all parties involved in influencing decisions that affect their collective quality of life. The second emphasizes shared leadership that cuts across the institutional and community fabric continuum [40,41]. Hence, many political pronouncements and research projects have underlined integrated urban governance as a key approach for more sustainable urban development in unplanned areas. Primarily, the advance enhances participation at the community level and fosters a 'bottom-up' approach as a core aspect of informal settlements' economic and social development [42,43]. However, integrated urban governance has been tackled only as a management approach that refers to horizontal integrations between policy sectors (different departments) and vertical intergovernmental integration (between different tiers of government) and beyond administrative boundaries. The approach not only cuts across issues in policy-making that transcend the boundaries of established policy fields; it also goes beyond traditional sectoral and discipline-oriented decision-making and implementation. That requires significant changes in research and administrative settings

to manage a wide range of social, behavioral, economic, and public health aspects and built environment qualities that reflect the complex nature of any informal context.

On the same line, Mahabir et al. [8] examined informal contexts through an interdisciplinary approach to ensure a more holistic and systematic assessment [8]. Considering the complexities of informal contexts and their multifaceted nature, their social and physical constructs are signified. From this perspective, motivations that lead people to live in slums and the potential of the physical location should be investigated to develop and explore appropriate policies to improve dwellers' well-being.

Even from the environmental dimension, slum upgrading interventions are perceived beyond the restrictive approach to technical and normative issues. Understanding the economic and political framework, including the complex relationships involved in the process, could contribute to better solutions for socio-environmental conflicts. Thus, concerning urban governance policies, adopting an interdisciplinary approach to intervention projects encourages the construction of a conciliating cross-sectoral approach that prioritizes social justice [10].

Case study research in different parts of the world has revealed that good practices in slum upgrading and urban development have commonly relied on an interdisciplinary approach to the issues linking environmental analysis, urbanism, and socioeconomic perspectives [9].

Based on the above, this paper used an interdisciplinary approach to provide key steps toward developing integrated urban governance agendas for unplanned settlements in Egypt. Concerning the multifaceted nature of the informal contexts, practicing an interdisciplinary approach in research examinations allows one to visualize the relations among different variables of complex problems related to informal urban settings, then build upon such links an integrative perspective to solutions.

Challenges in practices of upgrading the informal settlements in the Egyptian contexts demand a blended interdisciplinary/transdisciplinary approach. Interdisciplinary methodologies are the cognitive processes of critically evaluating varied disciplinary insights. At the same time, the transdisciplinary approach exchanges various ideas, perspectives, and experiences among all stakeholders to reach a common ground for community perspectives and generate a widespread integrative knowledge base to direct policies [44,45].

The blended interdisciplinary/transdisciplinary approach could also guide the integrative research process for examining aspects of informal urban contexts while enabling the key elements of managing informal settlements in the Egyptian contexts. Sharing visions among all professions, government, and stakeholders eliminates conflicts and confrontations that generate problems and complications in the planning and implementation phases of the upgrading projects. While developing a comprehensive understanding of problems and then generating alternative solutions, integrated urban governance agendas could be developed to improve the quality of life in unplanned areas within the Egyptian context.

### 2.3. Understanding the Mechanism of Informal Settlement Deformation

This paper acknowledges urban governance as a process encompassing formal and informal arrangements to impact aspects of sustainable urban development [46]. Accordingly, individual citizens and households of all groups have power over the formation process of the informal domain. Thus, delivering multiple socioeconomic interactions through time, the formation process is shaped by households' histories, which creatively adjust to the flowing and changing physical context. The process verifies that cities must be conceived from the bottom up by tracing complex mechanisms of human experiences that activate prevailing streets or urban public places [16,47]. It could be determined that bottom-up integrative approaches for examining informal settlement deformation have taken three main perspectives as follows.

### 2.3.1. The Socioeconomic Perspective

In the early 1970s, a new paradigm and urban policy were provided by John Turner, who argued that housing was best provided and managed by those who were to dwell in it rather than being centrally administered by the governmentHe showed that settlements

self-designed by local groups worked better as people were experts on their situations and should be given the "freedom to build." Within this framework, the government and the private sector act as enablers [48].

Following the same perspective, De Soto proposed the importance of land tenure formalization in his book, *The Mystery of Capital*, published in 2000 [49]. According to De Soto, the leading cause of poverty in developing countries is low-income people's continuing lack of access to formal property rights. An essential idea concerning land tenure validation is that granting property titles is often presented as a means to provide credit-constrained households with better access to mortgaged credit, thus stimulating investment in business activities. In this view, if low-income groups are to gain access to the benefits of capitalism, their assets must be registered and integrated into unified property systems. The effect of land tenure formalization has been verified in various fields of study worldwide.

However, several works of literature have addressed that most land titling programs have failed to achieve the benefits claimed by De Soto [50,51]. The negative aspects of land tenure formalization and socioeconomic difficulties in implementing formalization have been identified as critical challenges, particularly their relevance to the spatial attributes of settings in urban informal contexts.

### 2.3.2. The Morphological Perspective

Understanding the process of informal settlements' formation in our cities has also been studied with more focus on the built environment facet. Thus, the spatiality of informal morphologies has been explored, not only different forms of informality at the city and neighborhood scales but also dynamics of change at the microscale levels [52]. Within this perspective, several essential clues were underlined to realize the formation of informal settlements. For example, topographic conditions and the existing pathways determine the access network of informal settlements [53,54]. Bhatt and Rybczynski [55] have identified physical elements characterizing informal spaces, including housing extensions, workplaces, small shops, and streets, focusing on the role of trees, vehicles, and public structures. In addition, the relations between functional mix, accessibility, building density, street-life intensity, and public/private interfaces have been explored to understand better the microscale informal morphologies [52]. It has also been indicated how an informal structure can accommodate a mix of working and living [56]. Additionally, a vertical combination of formal and informal characteristics sheds light on how informal morphologies could emerge and grow within a legal structure [57].

There is also a temporal dimension to informal morphologies, which is about the processes of increased adaptation. Informal settlements have been considered 'complex adaptive assemblages' in which small-scale changes often emerge before large-scale spatial actions [58,59]. Moreover, it has been argued that informal settlements can be considered 'modern vernacular environments' since spontaneity is integral to the incremental processes of spatial change in both of them [60,61]. On the other hand, several informal morphologies emerged through incremental adaptations to formal contexts.

The incremental adaptation processes in urban environments incorporate a mix of formal and informal conditions [58]. In most cases, the growth of various informality forms may appear random and chaotic. At the same time, there is often an underlying logic to their continuation: proximity to job opportunities, transportation, and city centers. Thus, urban informality is not only sociopolitically challenging but also spatially problematic. Such an understanding raises questions about the capacities of urban governance strategies to enable or constrain the processes of self-organization and incremental adaptations.

### 2.3.3. The Sociophysical Perspective

Several studies have examined how informal settings are formed in developing cities by addressing the relationship between regulations/orders of social facets and aspects of land use and housing availability. It is argued that, in unplanned contexts, understanding

the exact nature and content of land tenure is not determined in the law paradigm but should primarily be perceived as a social relationship [48]. Thus, a complex set of informal and formal rules governs land use and ownership [50]. The processes through which households in the informal settlements access housing land are also structured and regulated by forms of social ordering. Their success in delivering large quantities of housing land is attributed to the social rightfulness they command, evidenced by the general acceptance and respect they enjoy from those whose relationships they regulate. Critical challenges have been identified when dealing with negative aspects of land tenure formalization, inseparable from understanding related social and economic aspects.

From this perspective, Salama's work has placed great importance on society and the formation of urban places while acknowledging an interdisciplinary framework focusing on the assessment and re-conceptualization of space as context [11,62]. Based upon Lefebvre's theory, Salama acknowledged the built environment of any context as the product formed through a triadic relationship of three different but related types of spaces: the conceived (imagined), the perceived (measured), and the lived (experienced). Figure 4 shows Lefebvre's key interpretation of the formation of settlements, and the pledge of citizens' rights to dignified housing and employment is echoed in the United Nations Sustainable Development Goals, in particular, SDG 11: a commitment to make cities and human settlements inclusive, safe, resilient, and sustainable [63–65].

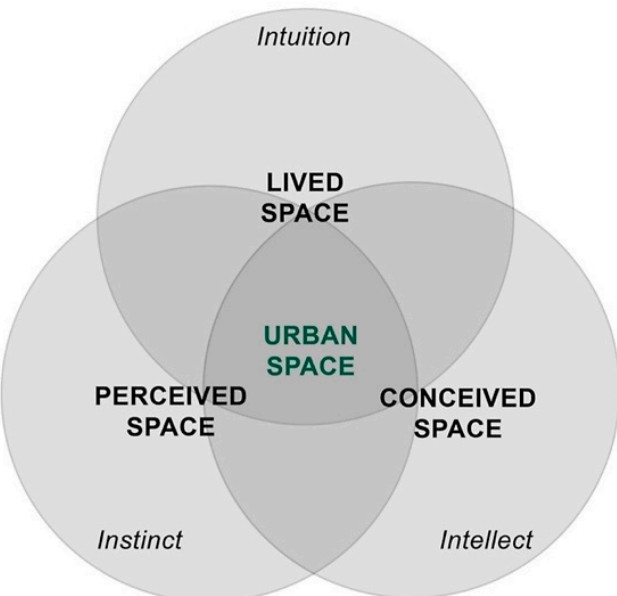

**Figure 4.** Lefebvre's key interpretation of the formation of spaces [11,66].

Salama argued that Lefebvre's framework facilitates understanding and investigating how the informal context is produced through three interrelated paths/stages:

- The imagined/conceived path by which stakeholders' settlement-relevant interventions are initiated;
- The measured/perceived path in which terms of economic and entrepreneurial practice lead actions to configure what is conceived;
- The experienced path in which the sociospatial context is lived.

The framework stresses the relationships between spatial and social practices as fundamentals to learn more about the production of informal areas in different contexts. Thus, spatial practice and social practice inevitably impact one another. In other words, these practices operate within a particular political process and, in turn, as part of a planning culture [62]. This embedded approach helps to understand the social dynamics in composing informal settlements and effectively align agendas around mutually desired aspects of urban governance.

Within the previous analogy, the conceived, perceived, and lived spaces in any context could be investigated within a sociospatial framework, where specific actors shape each space through complex variables involved. The process contributes to various parameters for examining and exploring urban performance in unplanned settlements. At the same time, qualitative and quantitative measures/parameters are associated with each of the three spaces.

Each of the three previous perspectives allows excellent insights into the bottom-up understanding approach for the formal contexts. They also support perceiving urban governance as a process incorporating formal and informal arrangements to impact aspects of sustainable urban development [46]. An integrative model for understanding the mechanism of informal settlement formation is introduced in Figure 5. The model accentuates the power of households over the formation process of the informal domains. Thus, delivering multiple socioeconomic interactions through time, the formation process is shaped by households' histories, which creatively adjust to the physical context's potential.

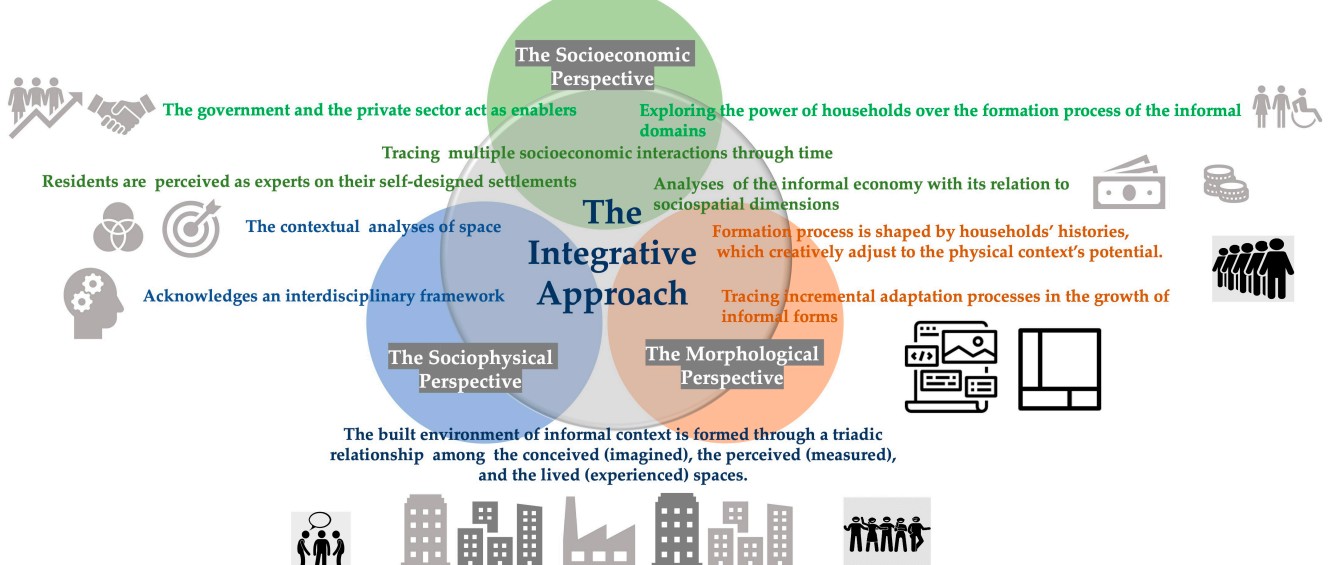

**Figure 5.** The integrative model for understanding the mechanism of informal settlement formation.

As the socioeconomic approach elaborates, the government and the private sector act as enablers [48]. Residents of the informal contexts are to be perceived as experts on their self-designed settlements. Consequently, exploring more analytical procedures to integrate the informal economy with its sociospatial dimensions is demanded to guide tenure formalization systems. The morphological perspective underlies the incremental adaptation processes in the growth of various informal forms allowed by a mix of formal and informal conditions [58]. Thus, urban informality is not only sociopolitically challenging but also spatially problematic. Therefore, the sociophysical perspective acknowledges an interdisciplinary framework that focuses on assessing and re-conceptualization space as context. It calls for multidimensional analytic procedures to the built environment of informal context as the product formed through a triadic relationship of three interrelated types of spaces: the conceived (imagined), the perceived (measured), and the lived (experienced) [11].

The next part is devoted to explaining how the mentioned integrative model could guide methods for overcoming the previously mentioned shortcomings of networking and the absence of an integrative knowledge base brought about by current urban governance practices in the Egyptian unplanned informal contexts. In parallel, the integrative model lays the foundations for establishing the blended interdisciplinary/transdisciplinary research process for examining aspects of informal urban contexts while enabling the key elements of managing informal settlements in the Egyptian contexts.

## 3. Materials and Methods

Concerning built environment design, understanding the sociospatial discourse of informal urban settlements could contribute to developing more effective sustainable strategies to improve quality of life. By encouraging deep community engagement through responsive methodologies, several integrated policies could be designed to facilitate a road map for informal settlements' urban governance. The following segment elaborates on the originated scheme to trace human experiences of the existence of any informal contexts.

### 3.1. Examining Informal Contexts—An Integrative Bottom-Up Perspective

Building on the previous perspective, an integrative bottom-up method could be identified for understanding the formation process of informal urban settlements. The method could be presented in a three-layer analytic framework shown in Table 1. Each framework's three layers signifies the socioeconomic/spatial phase in the formation process of any informal context, as explained below.

- The first layer spots the conceived space of the unplanned settlement formed by actors with specific visions about imagined goals (economic/social) related to a particular geographic area. Meanwhile, they can establish rules and norms to achieve those goals. This initial stage explains the mechanism of preliminary informal urban actions, where the availability of informal economic resources, community social and political powers, and physical potentials, together with the absence of formal urban policies, allow the generation of the conceived space.
- The second layer of the analytic framework concerns the perceived space of the unplanned settlement that may be agreed by the same actors of the conceived space or by other stakeholders with common/same interests. Within this stage, realistic actions are taken to construct the abstract spatial ideas conceived before and to establish the base map of the conceived space. For example, connecting areas of job opportunities, transportation routes, commercial services, and residential settings are commonly perceived spaces that govern the spatial layouts of informal settings. Hence, economic networks and sociocultural factors are key variables in creating the perceived space of informal context.
- The third layer tackles the lived space of unplanned settlement, which is formed by occupation of the perceived space by everyday users. Within this stage, actions/interactions of users/environment relations take place. The social and physical characteristics and economic dimensions, together with intrapersonal processes of dwellers, are integrated to formulate the lived space of an informal setting.

### 3.2. Value of the Integrative Method for Current Urban Governance Practices

The initiated framework analogy suggests a wide range of both qualitative and quantitative key measurable aspects to understand the process of generating informal context. In consideration of current practices of urban governance in the Egyptian informal domains, the integrative bottom-up method has numerous beneficial sides that lead to tracing the deformation stages of the unplanned domains and the actors involved in establishing informal contexts. It also overcomes three shortcomings, mentioned before, in current practices of upgrading informal contexts in Egypt as follows.



**Table 1.** The three layers of the analytic framework.

| Spaces | Key Factors/Variables Involved | Actors | Measurable Key Aspects | Methodology |
|---|---|---|---|---|
| **Conceived space**<br>- The dominant factor producing space in societies<br>- It is abstract and tactical<br>- The actual "representative of space" | - Economic power<br>- Physical potentials (existing land /form/topography)<br>- Backgrounds/ideologies<br>- Practices of official control and legalities | Conceptualized by decision-makers who are in a position to impose their personal notion of "order" on the concrete world such as investors/community leaders /contractors/architects and planners who work for the informal sectors | - Land uses /transformation<br>- Dominant building types/materials<br>- Dominant community powers/norms/rules<br>- Economic/base network | - Political economy analysis<br>- Analysis of government interests in upgrading the areas, together with municipal plans (if any)<br>- Interviews with key community decision-makers<br>- Analysis of informal governance models<br>- Analysis of historical maps (evolutionary maps) |
| **Perceived Space**<br>- The space of "spatial practice" where movement and interaction take place and networks are developed and materialized.<br>- It is a pragmatic, physical space encompassing flows of investment, workforce, and information, and this is where the conceived and lived spaces are construed | - Economic structure<br>- Physical availability/potentials<br>- Social structures<br>- culture norms | Perceived by<br>1- Investors<br>2- Community leaders<br>3- It includes both daily routines on an individual level and the networks that link places allocated for work, leisure, and "private" life | - Accessibility<br>- Connectivity + spatial mobility<br>- Buildings characteristics<br>- The general organizations of life support<br>- Social network<br>- Spatial configuration<br>- Issues to be raised/investigated: safety, health, poverty | - Socioeconomic network analysis<br>- Spatial analysis for the entire Informal settlement<br>- GIS analysis<br>- Mobility/connectivity mapping<br>- Interviews with key developers and investors |
| **Lived Space**<br>- The direct unconscious, relationship of human behavior to space; also known as "representational space"<br>-It is the most subjective space, involving the actual experience of individuals | Economic:<br>Family income<br>Physical: form, dimension, materials<br>Sociocultural background/<br>-lifestyle/traditions. | The everyday users including different types of residents: all actors in the "perceived space" and as a result of the "conceived space" | -Users' attitudes, preferences, satisfaction/environment behavior interactions including place identity, familiarities, safety, health, privacy, social interactions, territoriality<br>- Demographic information/housing characteristics | -Attitude surveys/questionnaires, Systematic observations Identification/attachment analysis and mapping (cognitive/behavioral) |

- The conceived space analogy paves the road to developing relationships and trust among formal and informal governance systems by acknowledging the community's key powers. It also allows recognizing the inter-relations of socioeconomic and sociophysical aspects of the informal contexts over time.
- The perceived space analogy suggests participatory procedures for the urban governance agendas by recognizing the multifunctionality and working mechanisms of informal economic networks together with sociocultural aspects in creating the base map of the conceived space.
- The lived space analogy suggests an integrative knowledge outcome. Actions/interactions of users/environment relations are the core of investigations. Thus, critical issues related to sustainable quality of life, such as place identity, safety, health, privacy, and social interaction needs, could be examined. Outcomes allow for the development of comprehensive urban governance agendas.

## 4. Results/Developing Integrative Urban Governance Agenda for Informal Settlements—A Road Map

As previously clarified, while handling the unplanned settlements in Egypt is a complex challenge, understanding socio/economic/health aspects and the interrelated physical mechanisms that lead to establishing informal urban communities is essential to overcome problems and challenges confronting urban governance agendas. An integrative approach is required to examine the complex, interrelated variables in producing the unplanned context. Blended interdisciplinary/transdisciplinary research methodologies allow the participation of academics and experts from different disciplines (sociology, economy, psychology, built environment design, public health) together with informal actors (inhabitants, commuters, and contractors). Desired outcomes are planned transparently and shared in terms of actors, roles, and collaborations to effectively measure all interrelated variables. A comprehensive database employing geographic systems is essential to allow context analysis processes. Thus, various investigations in different sectors (urban, economic, social), together with correlations and inter-relation analyses, generate inclusive concepts for developing the informal contexts.

Based on the above, two main compulsory aspects are to be elaborated to develop an integrative urban governance agenda, as follows.

The research team: This relies on the participation of an interdisciplinary team including both academic experts from different disciplines (sociology, economy, psychology, built environment design, public health), informal actors and stakeholders (inhabitants, commuters, and contractors) together with policymakers and authorized representatives.

Research schema: This comprises six joint layers as shown in Figure 6:

A. Theoretical, conceptual research models;
B. Analysis of economic/political/physical potentials of the informal context;
C. Socioeconomic network analysis (sociology/economy/urban planning experts involved);
D. Sociospatial analysis for the entire informal settlement;
E. Environment/behavior analysis;
F. Data interpretations and developing a project agenda for urban governance of the informal area under investigation.

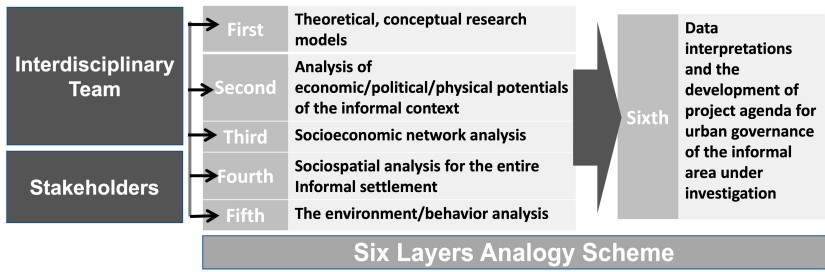

**Figure 6.** The Six-layer Analogy Scheme.

Each of the six interconnected layers has specific purposes, expected outcomes, and actors involved, as elaborated in the next part.

### 4.1. Theoretical and Conceptual Model Development

The first layer aims to establish an integrative theoretical base that allows an understanding of different perspectives and experiences regarding the problem of urban governance in informal settlements. Thus, the knowledge transfer process establishes a comprehensive theoretical model to build upon the following procedures. Participation of all academics and stakeholders is essential to develop common ground among them. Two main procedures could be followed:

- A theoretical analytic study to amalgamate knowledge from disciplines of urban/planning, sociology, psychology, public health, and economy. The ultimate goal is to develop comprehensive models that guide the field study tools, data analysis, and interpretations. Ultimately, models reflect all complex qualities of the informal settlements, including sociocultural, health, and economic, together with the physical attributes of the urban context.
- Both governmental and non-governmental stakeholders are invited to participate and share their visions concerning the integrative model introduced by academic experts.

Several techniques are employed to bring all parties together within the two procedures, such as mind mapping, conceptual mapping, and cognitive mapping. The developed conceptual models will be refined to design and guide integrative methods for the following layers.

### 4.2. Analysis of Economic/Political/Physical Potentials of the Informal Context

Investigating the *conceived space* in the informal area is the aim of the second layer. It could be achieved by graphing the origin of economic and political mechanisms in the unplanned area. Experts from economy/urban planning and GIS fields are to work together to scan agents of economy, land ownership, and history of community role models in the unplanned area.

In parallel, historical potentials of physical features in the area, such as vacant lands, accessibility to main roads, and adjacent land uses, is to be sketched.

### 4.3. Socioeconomic Network Analysis

In this layer, aspects of the *perceived space* are the investigation's main concern. The systems-based analysis allows an understanding of how the informal urban environments comprise various interrelated and interlinked systems (political, socioeconomic, infrastructural, etc.). Internal and external forces influence these in a specific urban setting. Sociology/economy/urban planning experts work together to examine available job opportunities, land ownership/rental costs of occupation, social benefits due to cheaper accommodation, and economic gains to the residents of the informal area.

### 4.4. Sociospatial Analysis for the Entire Informal Settlement

This layer is devoted to assembling geospatial and sociospatial information to support a broad-based integrative knowledge about the informal area. Thus, urban/architecture/sociology/public health/psychology/GIS experts are involved in two main tasks. First, understanding the physical properties of the informal area by providing a combination of GIS and spatial metrics. Second, providing multiscale analysis reflects the multiple scales of social systems that operate in the area under investigation, from an individual, household/family, to the community and the whole district.

In this layer of analyses, multiple integrative tools are designed to address issues related to the following five categories:

- The social database covers the head of households, the spouses, children, and other residents of housing units, the basic profile of the individual, age, gender, education, marital status, income, residence in the area, linkages to other geographical areas, education, employment, and skills.

- Satellites, maps, and photos initiate the spatial and physical databases. The data cover the informal setting's size, shape, distribution, density, and pattern. The spatial metrics show whether the structure of an informal settlement is regular or irregular, elongated or circular, dense or dispersed. In addition, road connectivity and accessibility and all location/outdoor spaces, dwellings, and building characteristics are to be measured.
- Public health data cover current health-related problems, conditions, and availability of health facilities, and community opinion on developing health disorders.
- Community stability and satisfaction data include crime rates, divorce rate/social consistency.
- Economic data cover an in-depth analysis of issues related to the migrants' social problems, their background, reasons for migration, duration of migration, their transition from and to other areas, mechanism of coping with the problems, issues related to willingness to pay for better living conditions and expectations of the government, and other members of urban society; the economic contributions of the people in slums; and the cost of alternative models of development of slum areas. The main tools and strategies suggested for data gathering in this layer of analysis are site visits using observation checklists, surveys using questionnaires, and numerous in-depth personal interviews with residents/everyday users of the informal settlements. Plan to sketch and photograph residents' houses, official maps, or aerial photos of the informal settlements.

Social, economic, physical, and health data are associated with the residential unit numbers, which link with the base map data. The output from these datasets then forms a series of thematic maps that portray spatial, socioeconomic, and health information.

### 4.5. The Environment/Behavior Analysis

This research methodology focuses on understanding the interrelation between the everyday users of the informal settings and the sociophysical characteristics of the informal settings. Mainly, investigations capture associations of everyday users perceptions/attitudes/preferences and style of life with characteristics of the built environment. Hence, sociology/public health/environmental psychology experts collaborate to conduct people-centered analysis methods. Within this level of analysis, issues of place attachment, familiarity, residents' satisfaction, home range, common diseases and health risks, and social diversity could be examined.

To investigate the place/space aspects of the everyday informal context, the following methods are suggested:

- As a sociophysical unit of environmental experience, the behavior setting analyses illustrate the interrelation between human actions/behavior and the physical attributes of places;
- The multiple sorting analyses identify everyday users' preferences, attitudes, and value systems toward current situations and future interventions;
- The visual response analyses explain how physical attributes of the informal setting are perceived and understood among everyday users.

The intended investigations explain how daily experiences of indoor/outdoor spaces in the informal context profoundly impact several aspects of the quality of urban life. On the one hand, the investigations call for innovative research tools that facilitate producing an integrative knowledge base to deal with the problems of informal settlements. On the other hand, they have immense value in deepening community participation, meanwhile allowing the professionals to understand the community's needs and attitudes toward the fundamental problems and the suggested solutions.

### 4.6. Data Interpretation and the Development of the Urban Governance Agenda

Within this layer, several intensive workshops are required among team members to interpret the research results, with two main tasks:

- Establishing analyses outcomes through results discussion per the literature, while the conceptual model for examining the unplanned contexts will be refined and elaborated;
- Then, scenarios for alternative frameworks/synergistic approaches for urban governance in the area under investigation will be suggested and introduced;
- Selected representatives from stakeholders are involved in evaluating and selecting the framework alternatives and finalizing the comprehensive urban governance agenda.

## 5. Discussion, Research Echoes, and Suppositions

This part is devoted to reflecting on the research analytic framework in terms of its purposes, methods, and outcomes initiated in the research scheme. As shown in Table 2 and Figure 7, two prominent notable points could be discussed as follows.

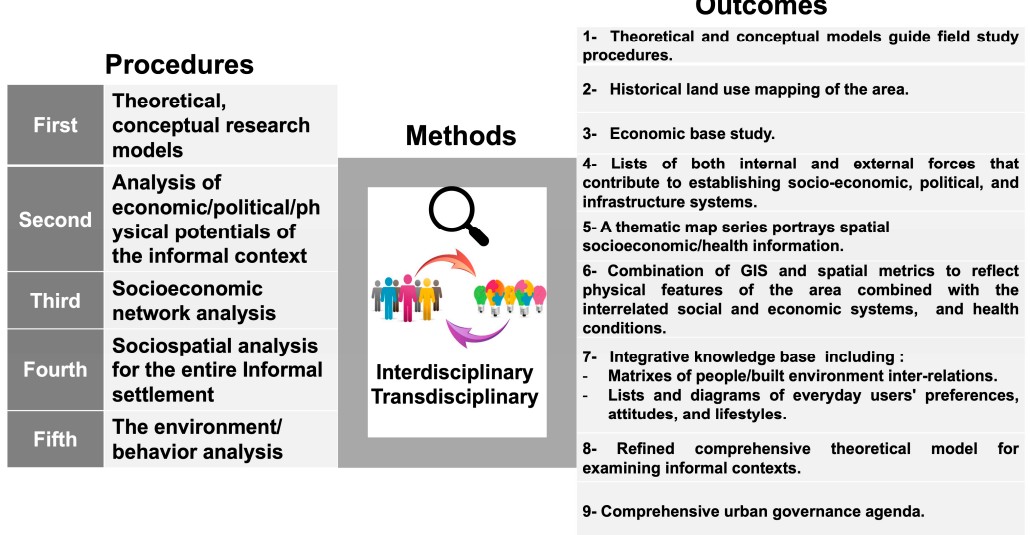

**Figure 7.** Procedures, methods, and outcomes of the research schema.

On the one hand, previous literature [8–10] challenged the multifaceted nature of informal contexts. The introduced six-layer analogy employs both interdisciplinary and transdisciplinary methodologies as core factors to examine informal contexts. Moving forward, structured, multidimensional analytic procedures are provided.

On the other hand, embracing the analogy model of conceived, perceived, and lived spaces in any context [11,32], this paper originates an integrative model for investigating the Egyptian informal contexts within a sociospatial framework, where specific actors shape each space through a complex of economic, politic, and other variables involved. The process contributes to various parameters for examining and exploring the urban performance of informal settings. In accordance, relations among different variables of complex informal problems could be visualized and then built upon in an integrative perspective to solutions.

While responding to current gaps in practices of upgrading the informal settlements in the Egyptian contexts [6,14,25,29,36], the introduced blended interdisciplinary/transdisciplinary methodology provides two core intended outcomes for dealing with problems of the informal settlements, which are the theoretical, conceptual models and the integrative urban governance agenda. In addition, nine underlying outcomes are identified and illustrated in Figure 7.

In addition, by tracking the conducted theoretical review, three core issues could be discussed concerning the six interconnected layers in the previous framework.

**Table 2.** Research schema composed of six joint layers.

| Layers | Main Purposes | Methods | Outcomes |
|---|---|---|---|
| **A-** Theoretical, conceptual research models | To establish an integrative theoretical base from different perspectives in regard to urban governance of informal contexts | ■ Interdisciplinary research strategy considering sociocultural, health, and economic factors, carried out by environment professions involved in problems of the informal urban context ■ Transdisciplinary research procedures to involve stakeholders (governmental and non-governmental). | 1- Theoretical and conceptual models guide the field study tools, data analysis, and interpretations of data |
| **B-** Analysis of both (informal/formal) urban potentials in the field study area (political/economic/physical analysis) | To explore socioeconomic potentials and the physical promises that facilitate informal context formation | ■ Blended interdisciplinary/transdisciplinary method. Experts from economy/urban planning and GIS fields, community leaders, and governmental stakeholders work together to scan agents of economy, land ownership, and the history of community role models in the unplanned area | 2- Historical land uses mapping of the area, including available vacant lands, accessibility to main roads, and adjacent services 3- Economic base study, including the availability of notable private business plans, human and natural resources |
| **C-** Socioeconomic network analysis (sociology/economy/urban planning experts involved) | To identify the interlinked systems (political, socioeconomic, and infrastructural) in the unplanned area | ■ Interdisciplinary method based upon outcomes of Layers A and B; systems-based analysis is to be conducted by social and economic professionals | 4- Lists of both internal and external forces that contribute to establishing socioeconomic, political, and infrastructure systems, including available job opportunities and attraction of investment prospects |
| **D-** Socioeconomic spatial analysis of the entire informal settlement | To assemble geospatial and sociospatial information to support a broad-based integrative knowledge about the informal area | ■ Blended interdisciplinary/transdisciplinary method. Urban/architecture/sociology/public health, psychology/GIS experts cooperate to understand the physical properties of the informal area and provide a multiscale analysis that reflects the multiple facets of social/economic systems that operate in the area | 5- Series of thematic maps portray spatial socioeconomic/health information 6- Combination of GIS and spatial metrics to reflect physical features of the informal area combined with the interrelated social and economic systems and health conditions that operate in the area |

**Table 2.** *Cont.*

| Layers | Main Purposes | Methods | Outcomes |
|---|---|---|---|
| **E-** The environment/ behavior analysis | To investigate the place/space aspects of the everyday informal context | ■ Blended interdisciplinary/transdisciplinary method; sociology/public health/ environmental psychology experts collaborate to use people-centered data-gathering tools and analysis methods<br>■ Tools are refined, developed, and applied to involve everyday users 'perceptions/attitudes/preferences and style of life with characteristics of the built environment | 7- Integrative knowledge base including: -Matrixes of the inter-relations between human actions/behavior and the physical attributes of places Lists and diagrams of everyday users' preferences, attitudes, perceptions, and value systems toward current situations and future interventions |
| **F-** Data interpretations and the development of project agenda for urban governance of the informal area under investigation | To introduce comprehensive urban governance agenda | ■ Blended interdisciplinary/transdisciplinary method<br>- All team members involved to discuss outcomes of Layers A, B, C, D, and E. Then, scenarios for alternative frameworks/synergistic approaches for urban governance to the area under investigation could be suggested<br>- Selected representatives from stakeholders are invited to evaluate the framework alternatives | 8- Refined comprehensive theoretical model to examine informal contexts 9- Comprehensive urban governance agenda |

First, the need for networking and gaps addressed in current practices of urban governance systems in Egypt [14,33,39,62]. The multifaceted nature of unplanned urban contexts is considered by merging interdisciplinary/transdisciplinary approaches for examining the informal contexts' interrelated social, economic, health, and physical dimensions along several accumulative layers of examinations and analyses. Thus, an integrative knowledge base and comprehensive urban governance agenda could be generated as outcomes. Meanwhile, the hierarchy of the analytic layers allows for establishing networking on two levels:

- Networking among the level of experts from different perspectives on public health, sociology, psychology, and built environment;
- Networking among experts, authorities, and residents involved in community-building processes to develop relationships and trust among formal and informal governance systems.

Second, the shortcomings of understanding informal contexts' formalization mechanisms in current urban governance systems practices. As previously clarified, while the unplanned settlements in Egypt are a complex challenge, understanding socioeconomic aspects and their related physical mechanisms that lead to establishing informal urban communities are essential to overcome challenges confronting urban governance agendas [15,16,18,50]. In this regard, the introduced research framework illustrates an integrative approach to examine the complex, interrelated variables involved in producing unplanned context based on the perceived, conceived, and lived space analogy [11,32,63]. A contextual analysis process for informal settlements could be achieved as follows:

- Layer B of the framework examines how conceived space is generated to form the unplanned context (formed by actors who have particular visions about imagined social and economic goals related to the physical potentials of the area);
- Layer C of the analytic framework focuses on the perceived space of the unplanned settlement by using base maps of informal urban governance to construct the abstract spatial ideas conceived before;
- Layers D and E tackle the lived space of unplanned settlement formed by occupying the perceived space by the everyday users. Within this stage, actions/interactions of users/environment relations take place. The social and physical characteristics and economic dimensions, together with the intrapersonal processes of humans, are integrated to formulate the lived space of an informal setting.

Third, regulating practices of integrative urban governance to develop only a management approach for the informal settlements. Integrated urban governance is one of many cross-cutting issues in policy-making that surpasses established policy sectors. It also goes beyond traditional sectorial decision-making to cover a wide range of social, behavioral, economic, and public health measures together with built environment variables that reflect the complex nature of informal contexts.

In this respect, the introduced framework, with its six layers, provides key steps towards heading integrated urban governance agendas for the unplanned areas through a blended interdisciplinary/transdisciplinary research methodology to scope the following key aspects:

Examining unplanned areas by visualizing the inter-relations among different variables of existing complex problems, then building upon such relations to gain an integrative perspective to solutions.

Examining aspects of informal urban contexts within shared visions and perspectives among experts, authorities, communities, and all stakeholders involved to develop a comprehensive understanding of problems and solutions.

Interdisciplinary/transdisciplinary methodologies employed in the current research framework enhance cognitive processes of critically evaluating disciplinary insights with community perspectives and create common ground to generate an integrative knowledge base for directing policies. On this base, integrated urban governance agendas and strategies could be developed to improve the quality of life in unplanned contexts.

## 6. Conclusions

As recovering quality of life in the informal contexts all over Egypt is addressed by Egypt's 2030 strategic plan for sustainable urban development, this paper finds that an innovative interdisciplinary/transdisciplinary investigative vision is the key approach to confronting complexity and the multiple facets of residents' quality of life in the unplanned areas.

The paper aimed to develop integrative urban governance strategies for informal settlements in Egypt by employing a comprehensive investigative framework based on a contextual perspective. It is contended that engaging in sociospatial, economic, and public health discourses of urban informal contexts is essential to developing efficient urban sustainable strategies. Meanwhile, informal practices of public intervention are claimed as sources of networked forms of power that work beyond the formal government and many other actors involved, including authoritarians and experts in social/economic/urban/health domains.

The theoretical research review revealed shortcomings in current practices. Lack of networking and gaps among urban governance systems, together with limitations in understanding formalization mechanisms of informal contexts, could contribute to inefficient urban governance agendas. While exploring three bottom-up perspectives to understand the formation of informal urban settings (socioeconomic, morphological, and sociophysical), the paper advocates an analytic framework of three layers of processes and mechanisms involved to establish urban informal contexts. Based upon the perceived/conceived and the everyday use space analogy, the research reached a correspondence framework to understand informal contexts' formation. A wide range of actors and essential measurable key aspects are defined for understanding the process of forming an unplanned context. It also suggests integrative knowledge outcomes to developing comprehensive urban governance approaches for informal contexts.

The work demands interdisciplinary teams from different disciplines (sociology, economy, psychology, built environment design, public health, and GIS experts), together with applying transdisciplinary methods for knowledge transfer among the professional team and stakeholders (inhabitants, commuters, investors, contractors), jointly with policymakers and authority representatives.

Conclusively, the integrative framework initiated in this paper is based on theoretical analytic methods. It is a contextual perspective deliberated as the significant procedure for developing integrative urban governance agendas. Finally, to improve aspects of quality of life in the unplanned settlements in Egypt, the integrative framework pertains to several informal contexts. Thus, different bottom-up agendas and experiences for more effective urban governance practices could be provided while refining interdisciplinary/transdisciplinary methods to accommodate a wide range of complexity addressed by varied informal settlements in Egypt.

**Author Contributions:** Conceptualization, E.E.N. and D.A.; methodology, E.E.N. and D.A.; validation, E.E.N. and D.A.; formal analysis, E.E.N. and D.A.; investigation, E.E.N. and D.A.; resources, E.E.N. and D.A.; data curation, E.E.N. and D.A.; writing—original draft preparation E.E.N. and D.A.; writing—review and editing, E.E.N. and D.A; visualization, E.E.N. and D.A.; All authors have read and agreed to the published version of the manuscript.

**Funding:** This research received no external funding.

**Data Availability Statement:** Not applicable.

**Conflicts of Interest:** The authors declare no conflict of interest.

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
