# Peer review of "The Inter/Transdisciplinary Framework for Urban Governance Intervention in the Egyptian Informal Settlements"

_buildings, doi:10.3390/buildings13020265_

Round 1

Reviewer 1 Report

Dear authors, thank you for choosing this journal.

The paper develops a topic of great interest worldwide considering also the continuous increase in population. The study , theoretically, analyzes well the state of the art , defining theoretical guidelines to the management of slums issues from an inter-transdisciplinary perspective.

In my opinion , however , even though theoretical the study should more refer to the case study where the research results are applied. So I suggest more expansion of section 3 and linking it to sections 4 , 5 and 6 where local context is no longer mentioned.
Suggest further additions as well:
- I would recommend citing the following article in the introduction that introduces a more holistic approach to the study of slums in order to truly understand them:

Ron Mahabir et Al. “The study of slums as social and physical constructs: challenges and emerging research opportunities”. Regional StudieS, Regional Science, 2016Vol. 3, no. 1, 399–419. https://doi.org/10.1080/21681376.2016.1229130

- Into the introduction I suggest citing data from "The 2020 Africa SDG Index and Dashboards Report" for Egypt, link follows:

https://countries.africasdgindex.org/#/egypt-arab-rep

- Paragraph 3 should include pictures, or maps to clarify the current status of slums in Egypt (Cairo)

- Clarify how study outcomes may be related to the local context

Author Response

Dear respectful reviewer,

We would like first to thank you for your valuable comments. Kindly find the following table with a point-to-point reply to your comments.

Thank you,

Comments from reviewer 1

Application of the modification

The theoretical study should refer more to the case study where the research results are applied. So I suggest more expansion of section 3 and linking it to sections 4, 5, and 6 where local context is no longer mentioned.

The paper has been restructured, and section 3 is now the literature review. It has been expanded and linked with other suggested sections.

I would recommend citing the following article in the introduction that introduces a more holistic approach to the study of slums in order to truly understand them:

Ron Mahabir et Al. “The study of slums as social and physical constructs: challenges and emerging research opportunities”. Regional Studies, Regional Science, 2016Vol. 3, no. 1, 399–419. https://doi.org/10.1080/21681376.2016.1229130

The suggested article has been cited in several parts of the paper, including the introduction and the literature review.

We appreciate the comment and the reference.

Into the introduction, I suggest citing data from "The 2020 Africa SDG Index and Dashboards Report" for Egypt, link follows:

https://countries.africasdgindex.org/#/egypt-arab-rep

The report has been cited in the introduction.

Paragraph 3 should include pictures or maps to clarify the current status of slums in Egypt (Cairo)

In the literature review under the section (Allocating Informal settlements in Egypt – current gaps ), We have included a map of greater Cairo's unplanned areas and a photo of a settlement on agricultural land.

Clarify how to study outcomes may be related to the local context

The discussion section has been updated and linked the results with the Egyptian context.

Reviewer 2 Report

Dear Authors,     

The paper entitled "The inter/trans-disciplinary Framework for Urban Governance Intervention in the Informal Settlements" aims to establish an integrative framework for the inter/trans-disciplinary study of urban governance for informal settlements in Egypt 

The research topic is highly relevant to the development of a sustainable city. This is because it raises very important issues related to the equitable development of cities, especially neighborhoods inhabited by the poorest population, or slums. In addition, the recently observed migration of people from the countryside to cities will exacerbate conflicts related to urban overpopulation.    

After reading the paper, I have comments and suggestions to improve the paper as follows:   

Structure of Article   

 I suggest improving the structure of the article according to the guidelines of the journal.   

New chapter numbering should be introduced   

1. introduction   

2. Literature review/Theoretical background   

3. Materials and Methods   

4. Results   

5. Disscusion  

Title: missing references, where was the research done?  

Abstract   

I suggest to correct to make it more readable.  I propose to correct the abstract according to the Journal "Sustainability". There is no information about the methods used and the results of the study are not presented.   

In Introduction   

In my opinion, it should be expanded with the following news: why was this study undertaken? What is most important here for the development of a smart city? What research has already been done so far, where? What conclusions are drawn from these studies. Is this article a continuation of those conclusions, or is it based on your own observations?  

At the end of the section should be the purpose of the research and the research questions.  

In Materials and Methods   

The authors present the research methods too generally, e.g., they write "Examination undertakings include Political economy analysis, Socio/economic network analysis, and Socio/Spatial analysis." without a thorough description of what the research will consist of?  

A diagram of the research procedure is missing.  

The Results [97-559].  

The results were presented and described in a good way, they are very interesting and important for the development of sustainable city. However, this part needs to be cleaned up. Some of the information should be moved to another chapter [97-221 Literature review. Similarly, subsections [222-371] provide more theoretical background than research results.    In Discussion     

This section should still answer the question: what tangible benefits has this study brought to the development of sustainable city?  It also important to describe the results of the paper in greater detail in this section. This would contribute to a high improvement of this paper. The authors should compare their project and results with results from similar conducted research on this topic from other parts of Europa and all around the world. 

Technical errors that need to be removed: 

Correct literature according to journal rules. 

All in all, I recommend this paper for publication in the Journal “Buildings after major changes. 

Kind regards,

Reviewer

Author Response

Dear respectful reviewer,

We would like first to thank you for your valuable comments. Kindly find the following table with a point-to-point reply to your comments.

Thank you,

Comments from reviewer 2

Application of the modification

Structure of Article   

 I suggest improving the structure of the article according to the guidelines of the journal.   

New chapter numbering should be introduced   

1. introduction   

2. Literature review/Theoretical background   

3. Materials and Methods   

4. Results   

5. Discussion  

The article's structure has been modified according to the reviewer's comments. The literature review as well.

Title: missing references, where was the research done?  

The paper title has been updated.

Abstract   

I suggest to correct to make it more readable.  I propose to correct the abstract according to the Journal "Sustainability". There is no information about the methods used and the results of the study are not presented.   

The abstract has been modified according to the reviewer's comment.

In Introduction   

In my opinion, it should be expanded with the following news: why was this study undertaken? What is most important here for the development of a smart city? What research has already been done so far, where? What conclusions are drawn from these studies. Is this article a continuation of those conclusions, or is it based on your own observations?  

The introduction has been modified to include the previous research that was done; however, the focus of our work is far from the development of a smart city. Furthermore, Cairo is not considered or planned to be a smart city. Therefore we cannot answer the importance of the development of the smart city.

At the end of the section should be the purpose of the research and the research questions.  

The introduction includes the research question, purpose, and aims.

Kindly check lines 68, 82, and 91 (in the file with track changes).

In Materials and Methods   

The authors present the research methods too generally, e.g., they write, "Examination undertakings include Political economy analysis, Socio/economic network analysis, and Socio/Spatial analysis." without a thorough description of what the research will consist of?  

A diagram of the research procedure is missing.  

The section on material and methods has been modified and updated.

The results were presented and described in a good way; they are very interesting and important for the development of sustainable city. However, this part needs to be cleaned up. Some information should be moved to another chapter [97-221 Literature review. Similarly, subsections [222-371] provide more theoretical background than research results.   

The structure and the literature review were modified according to the reviewer's comments and feedback.

In Discussion     

This section should still answer the question: what tangible benefits has this study brought to the development of sustainable city?  It also important to describe the results of the paper in greater detail in this section. This would contribute to a high improvement of this paper. The authors should compare their project and results with results from similar conducted research on this topic from other parts of Europa and all around the world. 

The discussion is modified. However, the slums and phenomena of informality in Egypt are difficult to be compared to the Europa context. Moreover, the previous papers examining inter/trans-disciplinary approaches dealing with the slums have been discussed in the paper as suggested by the reviewer.

Technical errors that need to be removed: 

Correct literature according to journal rules. 

The literature review was modified and updated.

Round 2

Reviewer 1 Report

Dear authors we appreciate the great effort produced in resetting the paper. The structure now makes clearer readability by adequately linking working method, discussion, and results. The references have been expanded and the Cairo map gives an overview of the issue helping the readers.

Reviewer 2 Report

The article has been revised according to the reviewer's suggestions. I make no further comments.